# Propeller Slipstream Effect on Aerodynamic Characteristics of Micro Air Vehicle at Low Reynolds Number

**Zhaolin Chen and Fan Yang \***

College of Aerospace Engineering, Nanjing University of Aeronautics and Astronautics, Nanjing 211106, China; zhaolin_chen@nuaa.edu.cn
\* Correspondence: fanyang@nuaa.edu.cn

**Abstract:** A numerical investigation on propeller-induced flow effects in tractor configurations on a Zimmerman wing-fuselage using the cambered thin airfoil is presented in this paper. The Reynolds number based on the mean aerodynamic chord was $1.3 \times 10^5$. Significant aerodynamic performance benefits could be found for a propeller in the tractor configuration. The numerical results showed that the propeller slipstream effect on the wings was highly dependent on the size of the propeller, and the major slipstream effect was working at 60% inboard wingspan, whereas less effects were observed towards the wingtip. The propeller slipstream increased the local angle of attack on the up-going blade side. This effect simultaneously augmented the section lift. The unsteady Reynolds-averaged Navier–Stokes (URANS) simulations helped to improve understanding of the interaction of the propeller wake and the wing-fuselage, which is an important aspect to guide the design of future efficient and controllable micro air vehicles. The results indicated that, in MAV designs, the slipstream from the propeller had a significant effect on the wing aerodynamics, regarding both performance and stability of the vehicle.

**Keywords:** Zimmerman wing planform; MAV; CFD; propeller slipstream effect; flow unsteadiness; low Reynolds number





## 1. Introduction

In recent years, with the rapid development of micro-electro-mechanical system (MEMS) technology, the concept of miniaturization of fixed wing [1], rotary wing [2], and flapping wing [3,4] designs have been rapidly carried out. With the implementation and promotion of the civil-military integration strategy worldwide, the application of micro air vehicle (MAV) technology in the civil field has made great progress. Substantial reduction in manufacturing cost was achieved due to the gradual maturity of MAV research and development of technologies, and such small flying vehicles have been widely used in various fields including agricultural plant protection, power line patrol, police and fire protection and search and rescue, damage assessment, surveillance, and reconnaissance [5,6]. However, the miniaturization of MAV designs still faces multiple challenges, including low Reynolds number aerodynamics, miniaturized structure design, low-speed propeller propulsion system, flight stability and control, micro remote control, and sensor devices, etc. The key issue for a MAV toward miniaturization in general is: airfoil performance deteriorates rapidly as the chord Reynolds number decreases below 100,000 [7]. This is because of the much stronger viscous effects including separation, transition, and reattachments, influencing the lifting surface performance. The formation of the laminar separation bubbles (LSBs), one of the pronounced phenomena which occurs in low Reynolds number aerodynamics, can significantly alter the boundary layer behavior and stall characteristics [8]. Early research focused on the aerodynamic characteristics of isolated wings, with more representative work such as Torres and Mueller [9], Mizoguchi et al. [10], Chen et al. [8,11], and Traub et al. [12]. However, the effects of propeller slipstream on aerodynamics and flow

physics have not yet been fully investigated and understood. Conventionally, it is common for the propulsion system to be considered separately from the wing and the airframe.

For MAVs, the interaction between the propeller and the aerodynamic surfaces can be much more significant due to the relative size of the propeller rotor disc to the MAV wing lifting surface, affecting its stability, performance, power consumption, noise, and overall endurance. Most of the MAVs are equipped with an electric motored propulsion system that contributes to the simplicity of operation. An obvious feature is that the aerodynamic efficiency of the wing control surface is significantly improved due to the flow separation that is prevented based on an extra momentum added to the flow from the propeller [13]. Most recently, propeller effects have been investigated on the vertical take-off and landing of MAVs [14], in which the authors found that the control surfaces became more effective as the slipstream strengthened. Therefore, the aerodynamic performance for the isolated wings is no longer applicable after a propeller is installed. The propeller swirling flow significantly modifies the surface pressure distribution and a considerable shifting of the center of pressure occurs, resulting in a change in overall pitching moment [15]. The swirling flow produced by the propeller also generates an additional yawing moment and a side slip can occur [16]. This is due to the propeller swirling flow effectively modifying the angle of attack of the downstream wing, thereby changing the wing's circulation [17]. On the other hand, the local angle of attack of the propeller blades is also modified by the upwash of the wing behind the propeller. The downward rotating blade increases the local angle of attack and increases the local lift and blade loading, which augments the thrust and torque on the blade.

The pitching moment coefficient ($C_m$), as an important characteristic, is often overlooked when considering airfoil performance. The flying wing configuration is normally desired for MAV designs and such a flying wing configuration must make a negative slope of the pitching moment curve ($C_{m,\alpha}$), and Cm at $\alpha = 0°$ ($C_{m,0}$) must be positive (+) (Nickel and Wohlfahrt, 1994) [18]. The requirements cannot be achieved by a traditional cambered airfoil, and hence a reflex camber must be designed which has a concave shape and locates near the trailing edge of the airfoil, shifting the pressure distribution aft and placing more upward force on the lower surface of the airfoil. However, the addition of reflex camber reduces the overall lift generation capabilities of the airfoil but is necessary for stable MAV flight. The limited amount of literature on cambered plate wings with a reflexed camber designed at trailing edge for low Reynolds numbers suggests that there is a need to expand the understanding of propeller-induced flow effects on reflex camber aerodynamic characteristics. To accomplish this goal, a computational study was performed to investigate the mutual aerodynamic influences between an MAV configuration and its tractored propeller. This is based on our previous studies on laminar separation bubbles [8] and planform effects [11] in a similar low Reynolds number range. The propeller–wing mutual interaction was studied and the comparison of wing aerodynamic characteristics of propeller on and off was conducted.

## 2. Computational Framework

### 2.1. Governing Equations and Solution Details

All simulations for this study solved Reynolds-averaged Navier–Stokes solutions. The flow equations were solved using the finite volume method, and the sliding mesh technique was applied for the interface between rotational and stationary domains. This led the computational domain to update the deformed mesh but change in time. Hence, the moving mesh integrated over a control volume was coupled with continuity and momentum equations, which can be written as follows:

$$\frac{\partial}{\partial t} \int_{V_c} \rho^f dV_c + \oint_S \rho^f (\upsilon - \upsilon_g) \cdot \mathbf{n} \, dS = 0 \tag{1}$$

$$\frac{\partial}{\partial t} \int_{V_c} \rho^f \upsilon dV_c + \oint_S \rho^f \upsilon (\upsilon - \upsilon_g) \cdot \mathbf{n} \, dS = - \oint_S \boldsymbol{\tau} \cdot \mathbf{n} \, dS - \oint_S p \cdot \mathbf{n} \, dS \tag{2}$$

where, $\rho^f$ is the fluid density, $\upsilon$ is the fluid velocity vector which is based on the Cartesian coordinates, $\upsilon_g$ is the moving grid velocity, $\tau$ is the molecular momentum transport tensor, p indicates the pressure gradients term, $V_c$ is the mesh cell control volume, $\mathbf{n}$ is the normal vector to the control volume surface, and S is the control volume surface area.

The $\gamma - \text{Re}_\theta$ transition model of Menter et al., 2006 [19], coupled the SST model with transport equations for the intermittency and $\text{Re}_\theta$ (i.e., momentum-thickness Reynolds number). The major improvement of this transition model was that it did not rely on nonlocal parameters, hence it was more suitable for modern CFD codes and complicated transitional-flow simulations. Further, a special modification to the intermittency was included to allow for separation-induced transition prediction. The SST $\gamma - \text{Re}_\theta$ has been used widely at both low [20] and high Reynolds numbers in aerodynamic applications. This paper adopted the $\gamma - \text{Re}_\theta$ transition model, which is proportional to the maximum strain-rate Reynolds number, i.e., presents the advantage of being a local property. The vorticity Reynolds number ($\text{Re}_v$) is defined as:

$$\text{Re}_v = \frac{\rho y^2}{\mu}\frac{\partial u}{\partial y} = \frac{\rho y^2}{\mu}\Omega \tag{3}$$

The $\Omega$ and y in Equation (3), are the vorticity and the wall normal distance, respectively, and the maximum value of $\text{Re}_v$ is dependent on the $\text{Re}_\theta$. The momentum-thickness Reynolds number transport equation was used to capture the nonlocal effect of freestream-turbulence intensity and pressure gradient at the boundary-layer edge, which indicates where transition onset occurs, and is defined as:

$$\frac{\partial\left(\rho\widetilde{R}e_{\theta t}\right)}{\partial t} + \frac{\partial\left(\rho U_j\widetilde{R}e_{\theta t}\right)}{\partial x_j} = P_{\theta t} + \frac{\partial}{\partial x_j}\left[\sigma_{\theta t}(\mu+\mu_t)\frac{\partial\widetilde{R}e_{\theta t}}{\partial x_j}\right] \tag{4}$$

The transport equation for intermittency is used to trigger the transition process ($\gamma > 0$) and is defined as:

$$\frac{\partial(\rho\gamma)}{\partial t} + \frac{\partial\left(\rho U_j\gamma\right)}{\partial x_j} = P_\gamma - E_\gamma + \frac{\partial}{\partial x_j}\left[\left(\mu+\frac{\mu_t}{\sigma_f}\right)\frac{\partial\gamma}{\partial x_j}\right] \tag{5}$$

However, when the boundary separates, the modification of intermittency to unity is defined as:

$$\gamma_{sep} = \min\left\{8\max\left[\left(\frac{\text{Re}_v}{2.193\text{Re}_{\theta c}}\right)-1,\,0\right]e^{-\left(\frac{R_T}{15}\right)^4},5\right\}F_{\theta t} \tag{6}$$

In the above equations, (5), $F_{length}$, and $\text{Re}_{\theta c}$ are two key functions, in which the former controls the transition extent and the later determines the onset of transition. The source term $P_\gamma$ is activated when the local strain-rate Reynolds number exceeds the local transition-onset criterion. The destruction source term $E_\gamma$ enables the relaminarization prediction when the transition-onset criterion is no longer satisfied, and vanishes in the fully turbulent regime. A complete description of the model is available in the article by Menter et al., 2006 [19]. This model has been used by numbers of researchers for low Re transitional flows. For example, a detailed study on two specific parameters ($F_{length}$ and $F_{onset}$ are used in the intermittency equation for controlling the length of transition region and onset location of transition, respectively) was shown by Suluksna [21]. Benyahia [22] conducted a validation study for the model for low Re number flows. According to the comparison between the numerical and experimental data, it was shown that the $\gamma - \text{Re}_\theta$ model accurately predicted the location and extent of the two-dimensional laminar separation bubble. Counsil [23] also studied the two-dimensional airfoils using the transition model, showing that the transition model was accurate in the preturbulent regions. A comparative study of four airfoils at low Reynolds numbers using the different transition modeling methods, namely, $\gamma - \text{Re}_\theta$ and $e^N$, was performed by Seyfert [24].

A blended first- and second-order accurate scheme was implemented for the spatial discretization, which switched from the latter to the former in regions of steep spatial gradients, based on the boundedness principle of Barth and Jespersen [25]. SIMPLEC algorithm [26] with a staggered grid and the second order-accurate central difference method was enforced, and such pressure–velocity coupling used coordinated under-relaxations for the momentum and pressure corrections to improve the convergence which was inherently slow in the original SIMPLE method. An implicit second-order quadratic backward approximation with an iterative procedure was realized for temporal discretization. The nonlinear coefficients were updated within each inner loop while the outer loop advanced the solution in time.

The entire computational domain consisted of a rotational zone containing the propeller and a stationary zone containing the MAV wing and fuselage, as shown in Figure 1. The multiple zones were connected with each other through non-conformal mesh [27]. The rotating domain contained a cylinder-type boundary, and the central axis which was coincident with the rotational axis of the propeller. The height and radius of the rotational domain were determined based on the propeller diameter ($D_{ia}$), indicating $2.55D_{ia}$ and $7.6D_{ia}$, respectively. The stationary domain, on the other hand, was set as a cubic block with a distance being roughly about $12.75D_{ia}$, $20.5D_{ia}$, and $12.75D_{ia}$ for upstream, downstream, and height, respectively; see Figure 1e,f. Structured mesh for both rotational and stational domain was generated, shown in Figure 1a,b. Mesh with high quality was considered and generated due to the importance of interpolation relationship between the interface surfaces, and a mapped mesh topology was proceeded between the interface boundaries. For the rotating domain, two O-topologies were created to cover the propeller and the center spinner segments. The O-grid included 10 cells normal to the propeller wall surface with a first cell distance of $2 \times 10^{-5}$ m. There were 30 grid points in propeller radial direction and 56 grid points in the circumferential direction. A cylindrical wake block was used to ensure a good resolution of the blade wakes and the tip vortices, which have a significant influence on the MAV domain. For the outer domain, a similar mesh topology as we showed in the validation case section was used. The total size of the mesh was about $8 \times 10^6$ nodes.

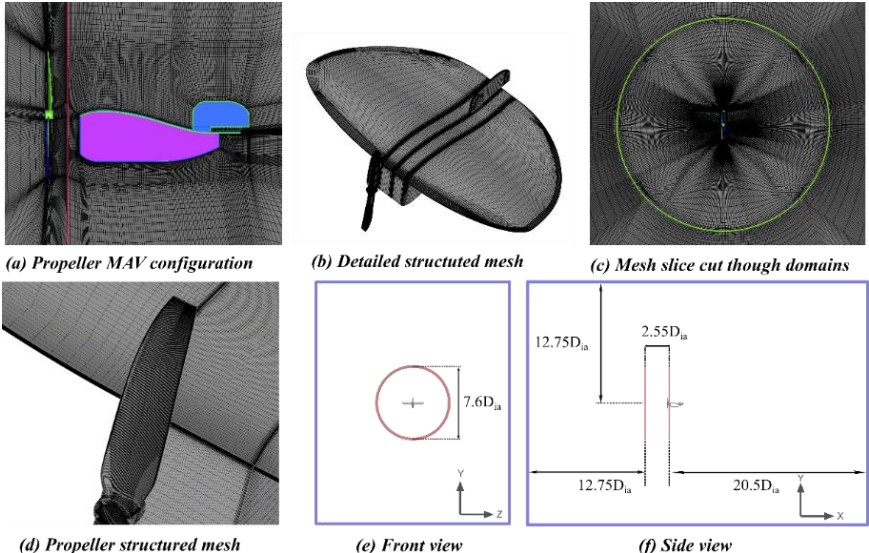

*(a) Propeller MAV configuration*  *(b) Detailed structuted mesh*  *(c) Mesh slice cut though domains*

*(d) Propeller structured mesh*  *(e) Front view*  *(f) Side view*

**Figure 1.** Non-conformal mesh interface for propeller MAV geometry, (**a**) propeller-MAV configuration, (**b**) detailed structured mesh for propeller-MAV, (**c**) Mesh slice cut though domains, (**d**) propeller structured mesh, (**e**) computational domain in front view, and (**f**) computational domain in side view.

The standard characteristic boundary conditions were applied on the farfield boundaries. For the case of low Reynolds number flow, total pressure, incoming freestream velocity, freestream turbulent intensity ($T_i$), and the turbulence length scale were imposed

at the inlet, whereas pressure was prescribed at the outlet boundary. Furthermore, at the solid wall, the non-slip boundary condition was applied. The turbulent kinetic energy was set to zero, and the pressure on the wall had zero normal gradients. High performance computing (HPC) with a parallel-processing implementation over 48 partitions and efficient message-passing interface between the partitions was adopted.

### 2.2. Specifications of Validation Cases

A three-dimensional Zimmerman wing planform was selected for the validation purposes, and the geometry was investigated by Torres and Mueller experimentally [9]. It has a zero camber and the aspect ratio (AR) is two, and the aerodynamic mean chord is 0.1725 m which gives a corresponding Reynolds number of 100,000. The model has a thickness-to-chord ratio of 1.96% and 5-to-1 elliptical leading- and trailing-edges. The aerodynamic center was assumed to be at the 25% point of the mean aerodynamic chord. To minimize the boundary condition effects, two different types of boundary settings were tested, and results were compared with the experiment data. Model 1 had an H-type mesh topology and the boundary conditions were set as velocity-inlet, pressure-outlet, and a non-slip wall boundary was applied on the wing. The farfield boundary condition was applied for the outer boundary. On the other hand, Model 2 was set as wing tunnel model boundaries. It has a velocity-inlet, pressure-outlet, and a wall boundary was applied for the wing and the outer domain faces, respectively (listed in Table 1). Model 2 was set as the wing tunnel model which had a contraction ratio of 20.6 to 1 and a rectangular inlet contraction cone. The freestream turbulence intensity in the test section was measured to be less than 0.05%. The test section was 182 cm long with a square cross-sectional area of 61 cm × 61 cm [9].

**Table 1.** Grid-sensitivity analysis for $\alpha = 4°$.

|  | Case | Grid Size | $C_L$ | $C_D$ |
|---|---|---|---|---|
| | G1 | 190 × 135 × 60 | 0.1744 | 0.01913 |
| Model 1 | G2 | 220 × 165 × 90 | 0.1775 | 0.02068 |
| | G3 | 250 × 195 × 120 | 0.1781 | 0.02052 |
| | G1 | 190 × 135 × 60 | 0.1899 | 0.01890 |
| Model 2 | G2 | 220 × 165 × 90 | 0.1900 | 0.01877 |
| | G3 | 250 × 195 × 120 | 0.1901 | 0.01879 |
| | Experiment [1] | NA | 0.1906 ± 0.02 | 0.0220 ± 0.003 |

An O-grid mesh topology was applied around the wing planform to capture the detailed flow field near the boundary layer for Model 1 and Model 2 configurations. Therefore, the grid points were concentrated near the wing planform edge, as shown in Figure 2d. For the wind tunnel setting case, a symmetric boundary condition was applied at the wing root plane, as shown in Figure 2c. The no-slip wall boundary condition was enforced on the wing surface. To minimize the farfield boundary condition effects, Model 1 had the domain which is set at $25\bar{c}$ upstream, $35\bar{c}$ downstream, and the upper and lower boundaries were placed at $25\bar{c}$ away from the airfoil leading-edge. However, Model 2 was set as wing tunnel model boundaries which had exactly same dimensions as the real wind tunnel.

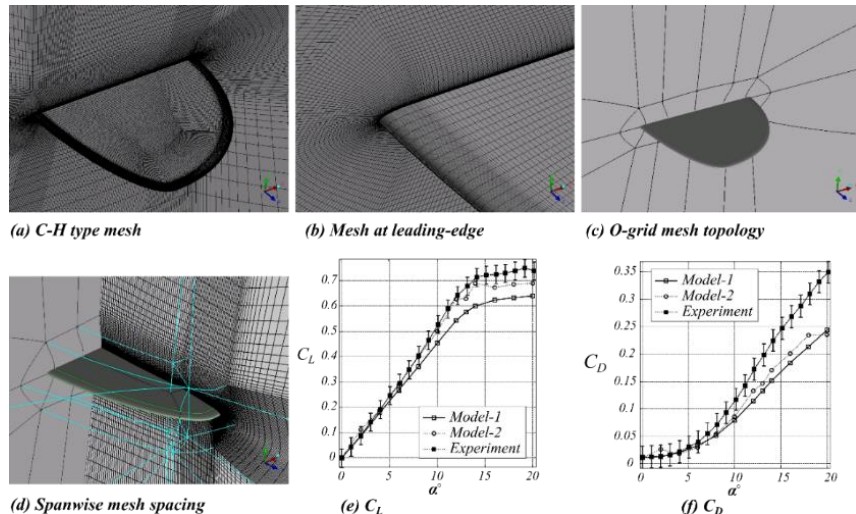

**Figure 2.** Mesh topology and aerodynamic coefficients for wing planform, (**a**) C-H mesh topology, (**b**) mesh at leading-edge, (**c**) O-grid mesh topology, (**d**) spanwise mesh spacing, (**e**) lift coefficient, (**f**) drag coefficient.

The mesh sensitivity was studied for the three-dimensional cases for $\alpha = 4°$ and the results are shown in Table 1. The baseline mesh (G1) had about 1.5 million mesh elements and model with a fine mesh (G2) had a doubled size of 3 million. Close to the wall, there were about 60 grid points within the boundary layer, and in the turbulent region, the $y^+$ value of the first cell distance was ensured to be in the order of O(1). The stretching ratio for the mesh was less than or equal to 1.2. Validation results are summarized in Table 1. The aerodynamic results comparisons, in Figure 2e,f, showed that Model 2 (wind tunnel settings) gave a better lift coefficient as compared with the experimental data. The results from Model 1 were close to the experimental data at low incidences but under-predicted at high incidences. The drag values showed a similar conclusion, showing potentially the stronger wall interference at higher incidences (Mueller [9]). From the mesh sensitivity study, the larger mesh size showed a reasonably better comparison than the coarse mesh. Therefore, for the propeller-wing-fuselage model presented later in this paper, a similar mesh topology, as shown for the validation case, was chosen and applied on the wing-fuselage part. The mesh for the propeller was integrated with the wing-fuselage mesh. The general topology is that an O-grid mesh was used around the propeller, and an H-type mesh was on the outer zone inside the rest of the rotating domain (details are shown in Figure 1).

### 2.3. Present Investigation Case

The model used in the current study was based on the flying wing MAV developed at the University of Sheffield [8,28]. This model is composed of a Zimmerman planform wing, a fuselage, a vertical stabilizer, and a propeller in a tractor configuration. The model is shown in Figure 3c with a coordinate system with x in the chordwise direction, y normal, and z spanwise. The wing has a mean aerodynamic chord of 0.221 m and an aspect ratio of 2.12. The freestream speed is 8.4 m/s, and the corresponding Reynolds number is $1.3 \times 10^5$.

The airfoil used in this investigation was a simple cambered thin airfoil, as shown in Figure 3b. It included both positive and negative cambers and the relevant parameters are listed in Table 2. The positive camber was designed to have better aerodynamic performance whereas the reflex camber was designed to maintain stable level flight, resulting in decreased flight times. Not only the static stability but also the dynamic stabilities (for control handling) were important for the design. Similar to Torres and Mueller [9], the lift center location was calculated from the normal force and pitching moment taken from the

$\bar{c}/4$ location of the wing. Although the location of lift center was close to $\bar{c}/4$ point at low angles of attack, it shifted towards the trailing edge as the incidence increased. The reason behind this is from both the increasing trailing edge separation and the strengthening wing tip vortices. In this investigation, the mean aerodynamic chord, $\bar{c}$, was used (Figure 3a).

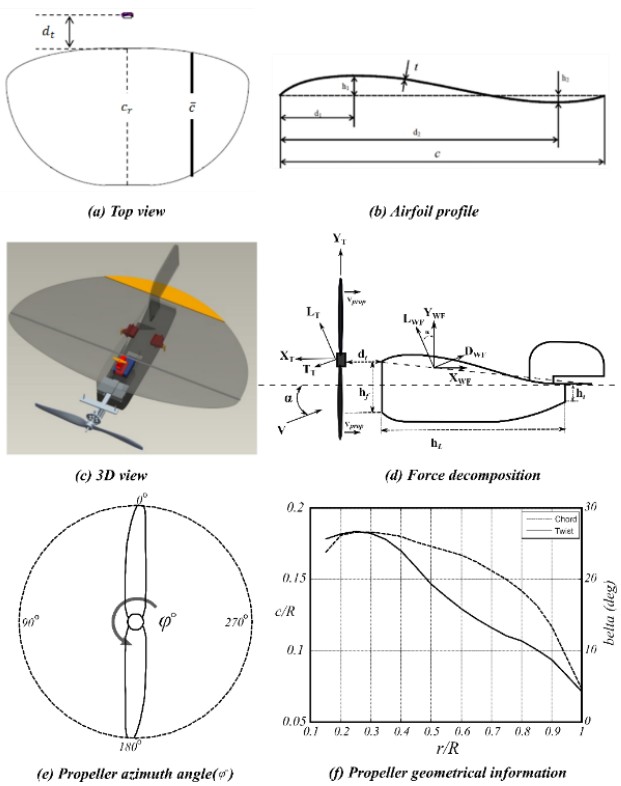

**Figure 3.** MAV and propeller geometrical descriptions, (**a**) Mav in top view, (**b**) airfoil profile, (**c**) MAV in 3D view, (**d**) aerodynamic force decomposition, (**e**) propeller azimuth angle, and (**f**) propeller geometrical information(chord and pitch angle distribution).

**Table 2.** Propeller location and fuselage specifications.

| MAV model | $d_t$ ($d_t/\bar{c}$) | $h_f$ ($h_f/\bar{c}$) | $h_L$ ($h_L/\bar{c}$) | $h_t$ ($h_t/\bar{c}$) | $t$ ($t/\bar{c}$) |
|---|---|---|---|---|---|
| | 0.068(31.7%) | 0.059(26.6%) | 0.216(97.6%) | 0.021(9.49%) | $2e^{-3}$(0.93%) |

The wing had a constant thickness of 2 mm and the shape of the cross section was rectangular with no sharp leading- and trailing-edge, and the airfoil is shown in Figure 3b. The camber was defined as maximum convex camber ($h_1/\bar{c}$), maximum concave and reflex camber ($h_2/\bar{c}$), maximum concave camber location ($d_1/\bar{c}$), and the maximum reflex camber location ($d_1/\bar{c}$); further details are shown in Ref. [11].

Considering the MAV configuration, the fuselage was a substantial component to accommodate the payload and propulsion devices, such as battery, motor, and servos, etc. Figure 3d shows the fuselage dimension for our model. It had a front height, $h_f$, of 0.059 m, rear height, $h_t$, of 0.021 m, and a total length $h_L$ = 0.216 m (Table 2). This design of the fuselage was dictated by the size and placement of the components. The fuselage layout affects the center of gravity margin and hence the static stability. For this purpose, the battery was designed to be movable to adjust the center of gravity. Another interesting point is how the fuselage affects the overall aerodynamics. Brion [29] simulated the fuselage and wing separately and the relevant aerodynamic forces are shown and discussed. However, the authors did not mention anything about the interaction between the wing and the fuselage. Ramamurti's [30] numerical results showed MAV with fuselage

reduced the lift-to-drag ratio dramatically and the drag for all configurations considered was nearly the same. The effects of the fuselage were also investigated in the present study.

The slow-flyer propeller was chosen for the current MAV study. The propeller was installed at a distance $d_t$ = 0.068 m ahead of the wing planform. It had a diameter of 8 in, the pitch was 4 in, and the hub diameter was 0.014 m. Figure 3e shows the propeller geometry and the blade azimuth angle. The propeller rotated in an anti-clockwise direction viewing from the front, and Figure 3f shows the propeller geometric characteristics. The aerodynamic balance determined a horizontal component X and a vertical component Y of the total force acted on the MAV model (Figure 3d). To obtain the overall lift and drag force on the model, horizontal and vertical components (i.e., X and Y components) were transferred into L and D components (i.e., based on the incoming freestream coordinates), as shown in Equation (7).

$$L = L_T + L_W, and\ D = (-T_T) + D_W$$
$$L_T = Y_T cos\alpha + X_T sin\alpha,\ and\ L_{WF} = Y_{WF} cos\alpha - X_{WF} sin\alpha \qquad (7)$$
$$T_T = -Y_T sin\alpha + X_T cos\alpha,\ and\ D_{WF} = Y_{WF} sin\alpha + X_{WF} cos\alpha$$

A time step of $1.5724 \times 10^{-5}$ s with 30 sub-iterations was applied for this study (is equivalent to 0.5 degrees per time step). To have reasonable numerical results, the $y^+$ value of the first grid point in the order of $O(1)$ was required. Figure 4 shows the numerical aerodynamic force coefficients versus the blade azimuth angle. It shows that periodic pulses were produced. This type of signature was found to be relatively independent of the advance ratio and appeared to be mainly associated with the local loading on the propeller itself.

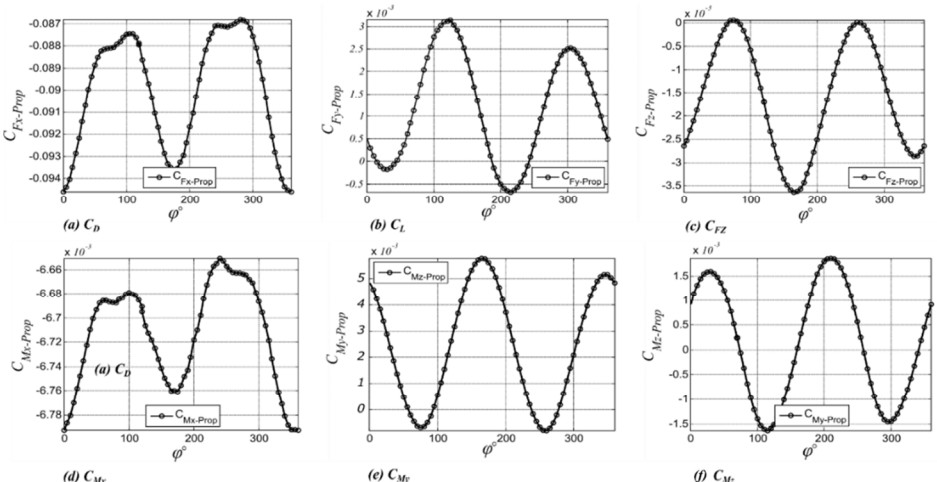

**Figure 4.** Propeller time history aerodynamic forces coefficient development, (**a**) drag coefficient, (**b**) lift coefficient, (**c**) side force coefficient, (**d**) rolling moment coefficient, (**e**) yawing moment coefficient, and (**f**) pitching moment coefficient.

A steady state flight condition is defined as one for which all motion variables remain constant with time relative to the body-fixed axis system XYZ. Mathematically speaking, steady state flight conditions are $\dot{\vec{V}}_p = 0$, and $\dot{\vec{\omega}} = 0$, and implies that MAV does not have any acceleration in any direction (i.e., $\frac{du}{dt} = \frac{dv}{dt} = \frac{dw}{dt} = 0$), and that the roll, yaw, and pitch rates are zero. Therefore, a propeller with rotational speed of 555 rad/s satisfied the steady state flight condition, as shown in Figure 5. Namely, the lift equaled to the weight and drag equaled to the propeller thrust.

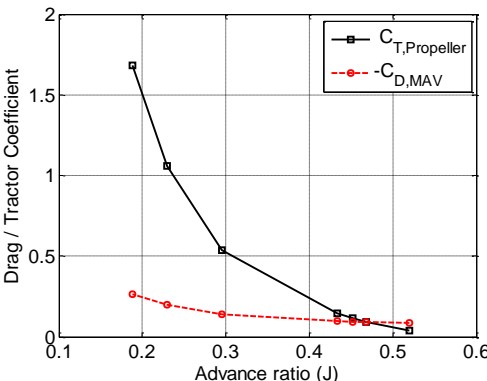

**Figure 5.** Drag and tractor coefficient with a function of advance ratio.

## 3. Results and Discussion

### 3.1. Propeller Slipstream Effects on Aerodynamic Performance

From previous investigations on both pusher and tractor propeller configurations, the latter was generally found to have higher aerodynamic efficiency and produce better control surface's effectiveness at landing and take-off. The MAV model considered here had a tractor configuration, and this was mainly to ensure the safety of the hand-throwing technique used on MAV take-off. Figure 6 illustrates the tractor propeller configuration effect on the left and right side of the Zimmerman wing. The propeller wake flow generated areas of upwash and downwash on the wing. Figure 6a shows the wing-airfoil aerodynamic loading in the upwash case where the propeller tangential velocity was greater than the downwash. The resultant force was tilted forward, and a localized wing thrust was created at this section. Figure 6b shows the aerodynamics of wing section under the propeller downwash. The wing downwash and the propeller tangential velocity acted together, resulting in a further reduction in the local angle of attack. The resultant force, F, shifted further backwards and resulted in an increased sectional drag.

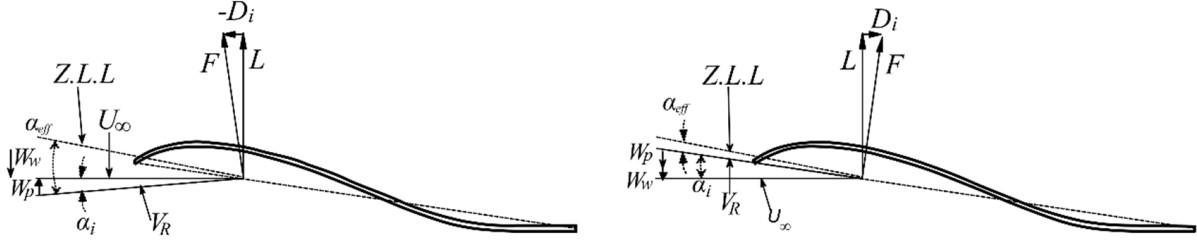

*(a) Wing section in propeller up-going blade*   *(b) Wing section in propeller down-going blade*

**Figure 6.** Propeller wing in tractor configuration, (**a**,**b**) wing local effective angle of attack due to propeller slipstream effect.

Figure 7a–f shows the aerodynamic coefficient of the propeller at a rotational speed of 555 rad/s. The forces and moments for different components clearly showed in the plots individually (propeller, wing-fuselage, and the vertical stabilizer). The lift coefficient in Figure 7a showed a linear variation for $\alpha < 6°$ and became nonlinear as the angle of attack was further increased. The maximum lift coefficient, $C_{L,max}$, was about 0.81 at the incidence of 12°. The propeller had negligible lift contribution at low angles of attack. The vertical stabilizer also showed a near zero contribution on the lift coefficient. At high angles of attack, the propeller had a significant contribution on the lift, which improved the maximum lift coefficient.

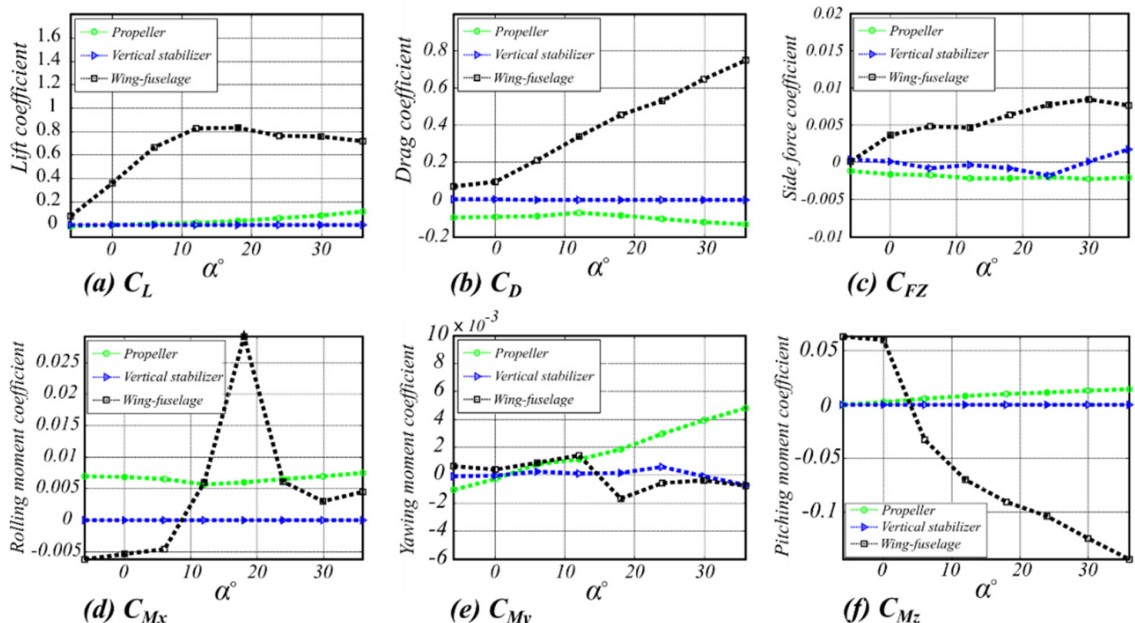

**Figure 7.** Averaged aerodynamic coefficients versus α at J = 0.468, (**a**) lift coefficient, (**b**) drag coefficient, (**c**) side force coefficient, (**d**) rolling moment coefficient, (**e**) pitching moment coefficient, and (**f**) yawing moment coefficient.

The drag coefficient, in Figure 7b, increased continuously with the angle of attack when the rotational speed was fixed at 555 rad/s. A nearly constant thrust was produced by the propeller with various angles of attack ($C_T$ is roughly around −0.09). The main drag force was formed from the wing-fuselage. The vertical stabilizer, however, showed a negligible drag contribution for the entire range of incidences.

Figure 7c shows the side force generated by the MAV model. In general, the MAV side force increased continuously as the angle of attack was increased. At 0°, the main side force coefficient was generated by the wing-fuselage, approximately 0.0037 (or 0.0143N). However, the propeller generated a negative side force coefficient of −0.0016 (or −0.0062N), and the side force from the vertical stabilizer was nearly zero at a 0° angle of attack. In general, the side force from the MAV had a small effect on the MAV as compared with the total weight.

The rolling, yawing, and pitching moments are shown in Figure 7d–f, respectively. A continuous increase of the rolling moment coefficient is shown in Figure 7d. At 0°, the overall rolling moment coefficient was 0.0014 (or 0.0024 Nm). A peak value was found at 18° mainly from the wing-fuselage part. The yawing moment coefficient, $C_{My}$, indicated that the MAV with a propeller spun clockwise as viewed from the rear and the moments caused a left yaw. However, it was negligible at low incidences around α = 0° with a value of approximately $9e^{-5}$. A linear increase in the propeller yawing moment can be found from the plot.

The pitching moment for analyzing the static longitudinal stability had a natural statically longitudinal stable region at α < 0° and positive contribution on the longitudinal stability side as the incidence increased. The pitching moment coefficient slope was closer to linear than that for the isolated wing-fuselage model in Ref. [11]. In contrast, the pitching moment slope, $C_{M\alpha}$, was about −0.0033 (for the linear section at incidence between 6° and 36°). The propeller showed a negative contribution on the statically longitudinal stability and a positive pitching moment slope can be identified in Figure 7f.

For the cruise condition, the lift force at zero degree of angle of attack was equal to the MAV weight and the thrust force could overcome the drag force, shown in Figure 7b. Figure 8a shows the locations of $C_p$ distribution on the Zimmerman wing at r/R = 1 (or z/b = ± 0.25). The $C_p$ distribution locations are indicated by the dash line for the up-going

and down-going blade sides. Both $C_p$ with and without slipstream at 0° flight condition are presented in Figure 8b–e: the instantaneous $C_p$ distribution is plotted with various azimuth angles (in solid line). The $C_p$ plots show that the positive lift force was generated at the positive camber area around x/c = 0.25 and the negative lift force was obtained from the negative camber, which was located near the trailing edge. The negative lift was also generated at near leading-edge area on the down-going blade side. The averaged $C_p$ at 2z/b = 0.5, in Figure 8f, shows that a similar amount of lift was produced by the isolated MAV model and the MAV propeller model. However, the negative lift generated at the reflex camber region by the MAV propeller model was quite significant, and more negative lift was produced due to the propeller slipstream effect. In comparison, the amount of negative lift almost doubled as compared with the isolated MAVs (dashed line). Therefore, less reflex camber could be used for the MAV with the propeller installed. The turning point is the point where the negative lift force started to occur, which shifted toward the leading edge slightly due to the propeller slipstream effect.

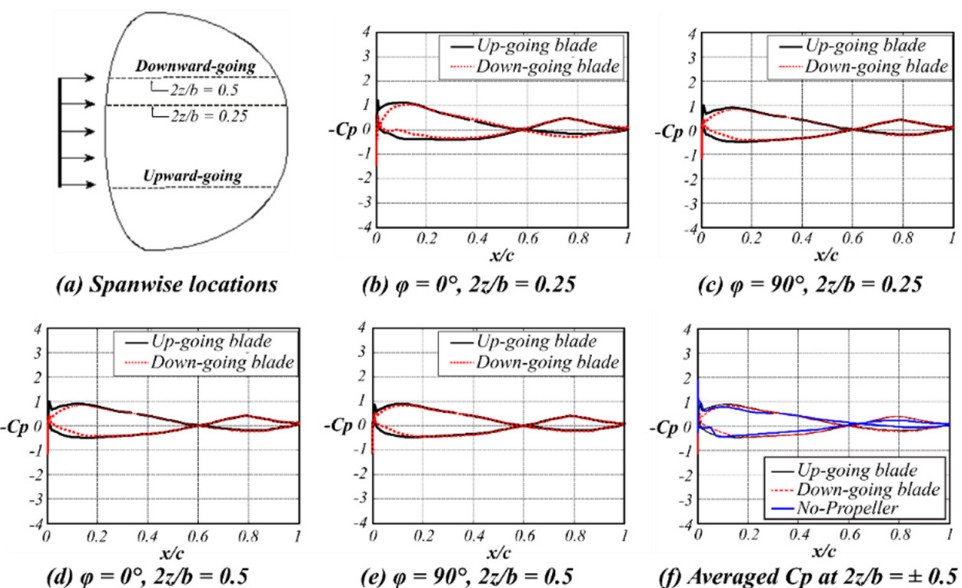

**Figure 8.** Instantaneous and averaged $C_p$ distributions at various spanwise location with $\alpha = 0°$ and $\omega = 555$ rad/s, (**a**) MAV wing spanwise locations, (**b**,**c**) wing spanwise location of *2z/b* = 0.25 and propeller zimuth angle of 0° and 90°, (**d**,**e**) wing spanwise location of *2z/b* = 0.5 and propeller zimuth angle of 0° and 90°, (**f**) pressure coefficient comparison between isolated wing and wing-MAV at spanwise location of $2z/b = \pm 0.5$.

Figure 9 shows the two-dimensional lift and drag coefficient distribution at various spanwise locations. The difference of the lift coefficients distribution for both wing sides is clearly shown. The drag coefficient, on the other hand, showed a higher value on the down-going blade and a slightly lowered value on the up-going blade. The possible explanation could be that in the up-going blade region, the propeller swirl counteracted the effects of the wing downwash such that the local angles of attack were increased. This effect simultaneously augmented the section lift and rotated the force vector forwards, which reduced the drag component at the section, as detailed in Figure 6. The sudden increase in lift or decrease in drag coefficient indicated the aerodynamic interaction on the wing and the fuselage from the propeller wake. In general, the major slipstream effect was working at 60% inboard wingspan, $2z/b \leq 0.6$, and less effects were observed toward the wingtip.

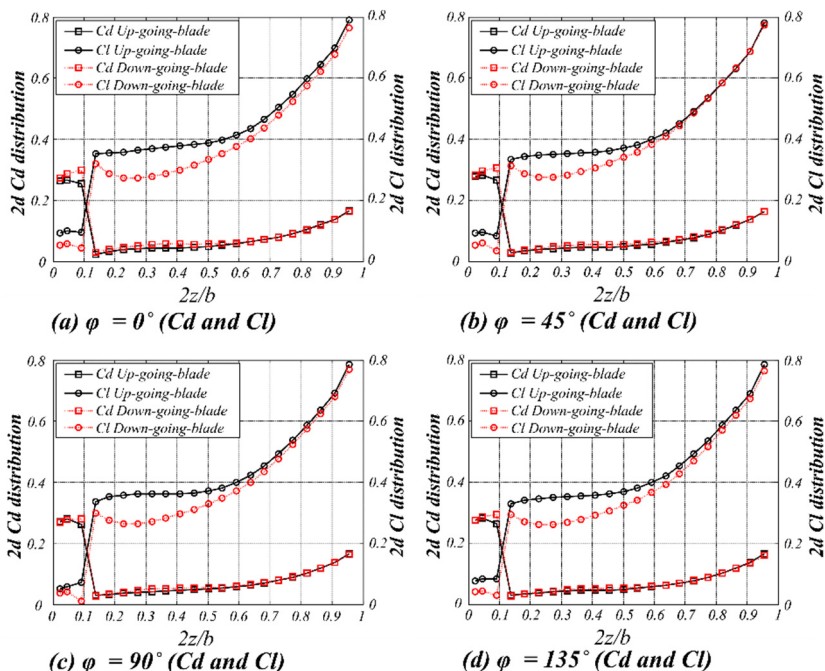

**Figure 9.** Instantaneous $C_l$, $C_d$ distribution at various propeller azimuth angles at $\omega = 555$ rad/s, (**a**–**d**) lift and drag coefficient distribution along the wing span with changing of propeller azimuth angle from $0°$ to $135°$.

In general, the propeller slipstream had a significant effect on both positive and negative lift. Identical negative lift was mainly generated at reflex camber area for both downward- and upward-going blade sides and asymmetrical forces were produced at the positive camber region due to stronger propeller slipstream, as shown clearly from the pressure distribution in Figure 8.

### 3.2. Propeller Slipstream Effect on the Flow Structure

Flow structures around and downstream of the propeller are presented and discussed in this section. Figure 10 shows the asymmetric flow structure of the MAV propeller model for the propeller at two different azimuth angles, $\varphi = 0°$ and $90°$, respectively, with $\alpha = 0°$. With the propeller slipstream, the wingtip vortex was reduced as compared with the MAV without the propeller slipstream, shown in the previous paper [11]. A clear asymmetric flow appeared, which followed the airfoil local inflow angle of attack evolution. Larger local angle of attack was formed on the upward-going blade side than that on the downward-going blade side, as shown in Figure 10e,g, respectively. Toward the wingtip, the local angles of attack on both sides were about the same. Details are shown in Figure 10f,h.

At $\varphi = 0°$, the leading edge separation bubble formed on the upper wing surface and it was located near the center plane region, as shown in Figure 10i. The low pressure region, as marked with dark blue color, can be clearly seen on the upper surface, indicating the asymmetric flow due to the propeller slipstream effects. On the lower wing surface, two separated and asymmetric bubbles formed underneath the wing near the leading edge. The smaller bubble formed on the upward-going blade side and the larger one on the downward-going blade side. The shape of the bubble at different spanwise locations can be found in Figure 10e–h. The areas with relative larger pressure force were on the lower surface at the up-going blade side (Figure 10d). This was due to the propeller tangential flow impinging on the lower wing surface. However, on the down-going blade side, the tangential flow impinged on the upper surface instead.

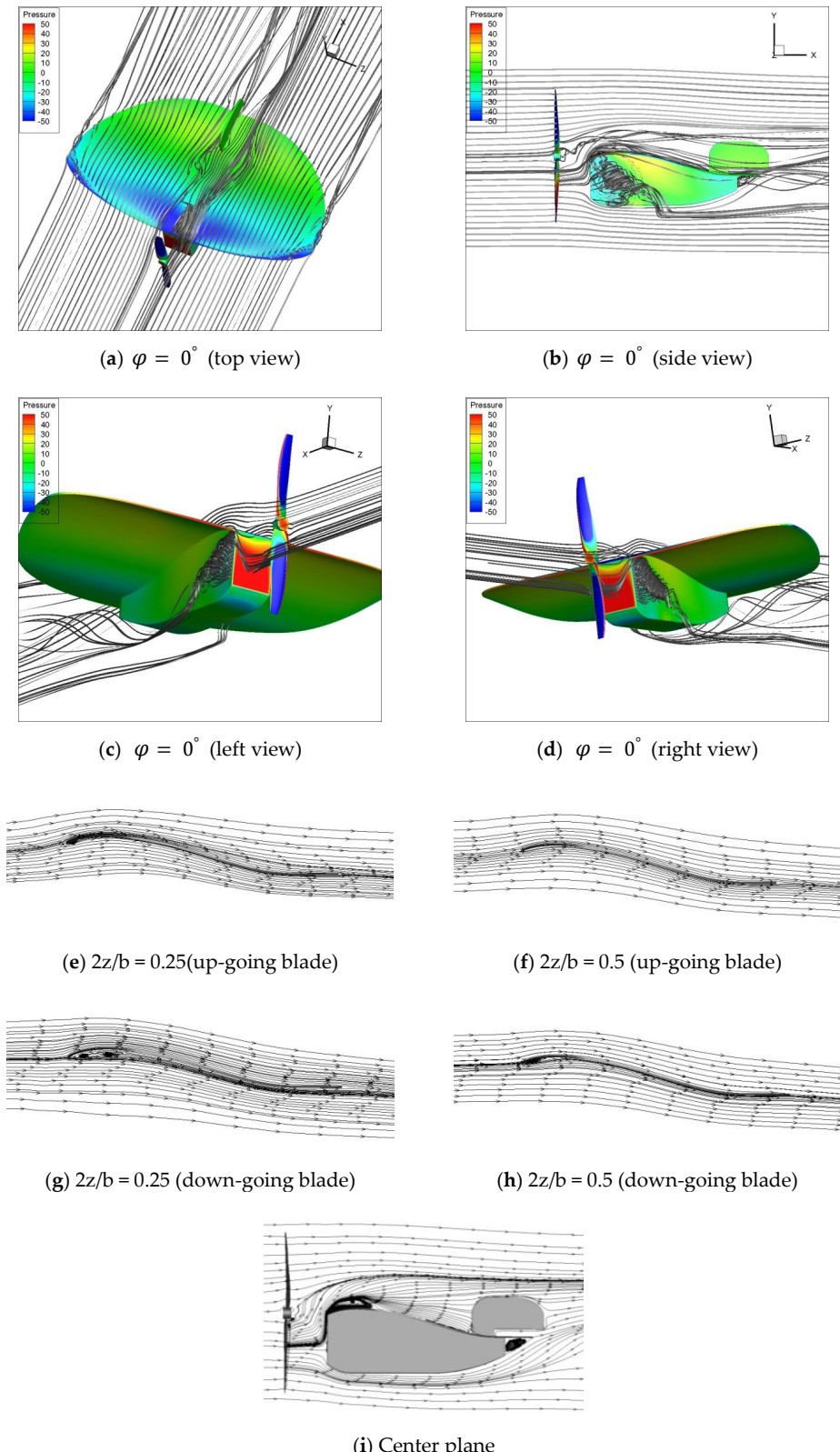

(**a**) $\varphi = 0^{\circ}$ (top view)

(**b**) $\varphi = 0^{\circ}$ (side view)

(**c**) $\varphi = 0^{\circ}$ (left view)

(**d**) $\varphi = 0^{\circ}$ (right view)

(**e**) 2z/b = 0.25(up-going blade)

(**f**) 2z/b = 0.5 (up-going blade)

(**g**) 2z/b = 0.25 (down-going blade)

(**h**) 2z/b = 0.5 (down-going blade)

(**i**) Center plane

**Figure 10.** Instantaneous flow structure around the MAV at $\alpha = 0^{\circ}$, $\varphi = 0^{\circ}$, (**a–d**) flow field around MAV, (**e**,**f**) airfoil sectional flow structure with up-going blade at spanwise location of 2z/b = 0.25 and 0.5, (**g**,**h**) airfoil sectional flow structure with down-going blade at spanwise location of 2z/b = 0.25 and 0.5, (**i**) flow structure at propeller's azimuth angle $\varphi = 0^{\circ}$.

## 4. Conclusions

The effect of a tractor propeller on the flow over a Zimmerman wing-fuselage model was investigated. The MAV propeller configuration was investigated with various advance ratios, and the equivalent rotational speed of 555 rad/s for the steady state condition was designed and the results were analyzed. Strong unsteady fluctuations were found in the slipstream region, showing development of periodic aerodynamic forces and moments. The sinusoidal fluctuation resulted in a large impact on the wing-fuselage aerodynamics. A severe mutual influence on the flow structure between the propeller and the MAV was observed as the angle of attack varied.

With the presence of the propeller, periodic aerodynamic force and moment on wing-fuselage-propeller configuration was observed. The contribution on the lift from the propeller increased as the angle of attack increased. While the propeller thrust and the drag force from the vertical stabilizer remained nearly constant for the incidence range, the drag from the wing-fuselage increased dramatically as the angle of attack increased.

The propeller slipstream effects on two-dimensional lift and drag coefficient distribution at various spanwise locations were clearly shown. The drag coefficient showed a higher value on the down-going blade and a slightly lowered value on the up-going blade. The possible reason could be that in the up-going blade region, the propeller swirl counteracted the effects of the wing downwash such that the local angles of attack were increased. This effect simultaneously augmented the section lift and rotated the force vector forwards, which reduced the drag component at the section. The sudden increase in lift or decrease in drag coefficient indicated the aerodynamic interaction on the wing and the fuselage from the propeller wake. In general, the major slipstream effect was working at 60% inboard wingspan, and less propeller slipstream effects were observed toward the wingtip.

The propeller slipstream showed a significant effect on both positive (for positive camber) and negative (for reflex camber) lift. The pressure distribution comparison between the isolated wing planform and propeller-installed model indicated that the down force caused by reflex camber due to the propeller slipstream effect was significant. In other words, the reflex camber of the wing needs to be reduced to achieve better aerodynamic performances for wing-propeller configuration.

**Author Contributions:** Conceptualization, methodology, and Writing—Original draft preparation, Z.C.; validation, F.Y. All authors have read and agreed to the published version of the manuscript.

**Funding:** This research was sponsored by the National Natural Science Foundation of China under Grant No. 11672132, 11672133, 12002161, and 12032011.

**Institutional Review Board Statement:** Not applicable.

**Informed Consent Statement:** Not applicable.

**Data Availability Statement:** Not applicable.

**Acknowledgments:** The authors wish to thank Gabriel Torres and Thomas J Mueller of University of Notre Dame for providing the wind tunnel data.

**Conflicts of Interest:** The authors declare no conflict of interest.

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
