# Peer review of "Propeller Slipstream Effect on Aerodynamic Characteristics of Micro Air Vehicle at Low Reynolds Number"

_applsci, doi:10.3390/app12084092_

Round 1

Reviewer 1 Report

Comments are included in the annexed word document

Type of manuscript: Article
Title: Propeller Slipstream Effect on Aerodynamic Characteristics of Micro
Air Vehicle at Low Reynolds Number
Journal: Applied Sciences

Comments

Figures and Tables are not properly referred throughout the complete paper. There are no references, instead there is “Error! Reference source not found…”.

The detailed schematics of stationary domain should be given in Section 1.1 to offer intuitive understanding for readers.

Figure 1 should be more detailed to better explain the mesh domain. Height and radius dimensions of the rotational domain should be added.

In Section 1, the subsections are not properly referred. In Subsection “Specifications of validation cases”, a three-dimensional Zimmerman wing planform was selected for the validation purposes, but in Figure 2 there is the half of the wing. It should be stated in the text.

In Subsection “Specifications of validation cases”, although the Model 2 is set as the wind tunnel domain which can be found from the Ref [32], more details should be added to the text or in a Figure to facilitate the read.

In Subsection “Specifications of validation cases”, the statment “Model 2 gives a better lift coefficient as compared with the experimental data” appears not to be adjusted with the results in Figure 2(e).

In Subsection “Specifications of validation cases”, the dimensions of the fine and coarse mesh for the Model 1 and 2 are missing in the text. It should be added.

In Subsection “Present Investigation Case”, the symbol D is referred to the drag force, but in Subsection 1.1 it was defined as the diameter of the propeller. It should be clarified.

 In Subsection “Present Investigation Case”, the statement “The peak values are found when the propeller at its horizontal locations, i.e.  and ” explain that the propeller azimuth angle is defined by  while in the Figure 4 there is other angle. It must be clarified.

The Section Results and Discussion is not properly referred. In this section, the parameter J stated in Figure 6 is not mentioned in the text. Could you please explain this?

In Figure 6 (a), “the propeller has a significant contribution on the lift’’, could it please be explained better? From what angle of attack and how much increase in lift would it produce in the vehicle?

The spanwise locations should be defined in the scheme of Figure 7 (a) in order to explain the results better.  

Author Response

We have made our response to the reviser's comments. Please check the attachment

Reviewer 2 Report

See attachment file

Author Response

(The authors gave the same response as above.)

Reviewer 3 Report

The article is a good subject but written at a middle level.
However, there are some points that need to be corrected, so I recommend a major revision of the article.

The title of the article completely corresponds to it.
The main comments to the article:
  • The design of the article should be updated, there are many questions (there is no affiliation, no citations in the article, etc.).
  • Due to the lack of citations in the article, it is not clear where the author's own results begin. All figures and graphs taken from other articles should have citations to the source.
  • The formatting of the equations is very strange and needs to be corrected.
  • Numbers should be in regular format and not in italics, which is found in the text and figures.
  • There are many questions regarding the presentation of results. The authors talk about increasing the lifting force when changing a certain angle. However, the article does not contain any graphs comparing the effect of this angle on the lifting force, but only on the coefficients. Therefore, it is necessary to provide such results to numerically understand the impact of the proposed solution.
  • In conclusion, the authors use the notation, this should be corrected. In addition, the conclusion should be clearer from the presentation of specific numerical results.

Author Response

(The authors gave the same response as above.)

Round 2

Reviewer 1 Report

None.

Author Response

The article is revised and updated, please find the cover letter for the summary of modifications.

Reviewer 2 Report

Table 1 shows a monotonic convergence on the lift coefficient for both models, while this doesn't happen for the drag coefficient. Could you please provide an explanation?

Please update the table 1 in the revised document with the third grid level too.

Author Response

(The authors gave the same response as above.)

Reviewer 3 Report

The authors have significantly improved the article and corrected many according to the comments. The only thing left to be corrected is the conclusion, where there should be more numbers on the results obtained. So I recommend a minor revision of the article.

Author Response

(The authors gave the same response as above.)
